# Review of Chemical Sensors for Hydrogen Sulfide Detection in Organisms and Living Cells

**DOI:** 10.3390/s23063316

**Published:** 2023-03-21

**Authors:** Mengjie Yang, Yong Zhou, Ke Wang, Chunfeng Luo, Mingna Xie, Xiang Shi, Xiaogang Lin

**Affiliations:** Key Laboratory of Optoelectronic Technology and Systems of Ministry of Education of China, Chongqing University, Chongqing 400044, China

**Keywords:** hydrogen sulfide, gasotransmitter, chemical sensors, organisms, living cells

## Abstract

As the third gasotransmitter, hydrogen sulfide (H_2_S) is involved in a variety of physiological and pathological processes wherein abnormal levels of H_2_S indicate various diseases. Therefore, an efficient and reliable monitoring of H_2_S concentration in organisms and living cells is of great significance. Of diverse detection technologies, electrochemical sensors possess the unique advantages of miniaturization, fast detection, and high sensitivity, while the fluorescent and colorimetric ones exhibit exclusive visualization. All these chemical sensors are expected to be leveraged for H_2_S detection in organisms and living cells, thus offering promising options for wearable devices. In this paper, the chemical sensors used to detect H_2_S in the last 10 years are reviewed based on the different properties (metal affinity, reducibility, and nucleophilicity) of H_2_S, simultaneously summarizing the detection materials, methods, linear range, detection limits, selectivity, etc. Meanwhile, the existing problems of such sensors and possible solutions are put forward. This review indicates that these types of chemical sensors competently serve as specific, accurate, highly selective, and sensitive sensor platforms for H_2_S detection in organisms and living cells.

## 1. Introduction

Hydrogen sulfide (H_2_S), often regarded as an environmental pollutant, is a colorless, flammable, and toxic gas with a distinct rotten egg smell [1]. In recent years, extensive studies have revealed that H_2_S is the third gasotransmitter following nitric oxide (NO) and carbon monoxide (CO) [2,3]. H_2_S is endogenously generated by three principal enzymes: cystathionine-β-synthase (CBS), cystathionine-γ-lyase (CSE), and 3-mercaptopyruvate sulfurtransferase (MST) [4]. Because of its various functions in physiological and pathological processes, H_2_S detection has become a research hotspot in the field of biology [5]. Currently, H_2_S is utilized in immune response, signal transduction, and energy production. Endogenous H_2_S mediates a wide range of physiological functions in the cardiovascular, neuronal, immune, endocrine, and gastrointestinal systems [6] in the aspects of the regulation of blood pressure, neurotransmission, anti-inflammatory effects, vasodilation, antioxidation, and apoptosis [7]. For example, the physiological concentration of H_2_S in blood lies in the range of 10–100 μM, and a close correlation exists between H_2_S levels in serum and disease states [8]. Additionally, changes in levels of H_2_S in the body have been linked to a variety of diseases, such as Alzheimer’s disease, Down syndrome, diabetes, cirrhosis of the liver, and even cancer [9]. Recent studies have unveiled that H_2_S levels are also closely related to asthma and chronic obstructive pulmonary disease (COPD) [10]. Due to the complex environment of organisms, the detection of H_2_S may be affected by other competitive analytes [11]; common competitive analytes in organisms and living cells are shown in Figure 1. In order to screen and monitor H_2_S levels in organisms and living cells and prevent health effects, a reliable, sensitive, and highly selective analytical technique is urgently needed.

At present, H_2_S detection methods such as electrochemistry, gas chromatography, high-performance liquid chromatography, inductively coupled plasma-atomic emission, UV-visible absorption spectrometry, and fluorescence spectrometry have been developed [12]. Of these, electrochemical sensors possess the unique advantages of high sensitivity, easy miniaturization, quantitative feasibility, low cost, simple operation, etc. [13]. At the same time, fluorescence and colorimetric chemical sensors exhibit the advantages of low cost, visualization, and being non-invasive, which bestow them with an irreplaceable role in the real-time detection of H_2_S in vivo [14,15,16]. These chemical sensors usually rely on three main properties, including metal affinity, reducibility, and nucleophilicity [17].

The principle of a chemical sensor for detecting H_2_S is shown in Figure 2. When H_2_S is in contact with test materials, corresponding chemical reactions occur with test materials based on different properties of H_2_S. Generally speaking, in the electrochemical sensor, the resistance of the electrode changes upon the reaction between test materials and H_2_S so that the electric current and other parameters change. After signal processing, the purpose of quantitative detection of H_2_S can be achieved. In the fluorescence and colorimetric chemical sensors, different concentrations of H_2_S after chemical reaction produce different intensity of fluorescence or show different colors.

In this review, our focus is on the recent decade of chemical sensors for the detection of H_2_S. A search was conducted in the Web of Science database using the keywords “hydrogen sulfide”, “gasotransmitter”, “chemical sensor” and “organism” or “living cells”, and summarized the relevant literature from 2012 to 2022. We classified the articles according to the different properties of H_2_S. The detection materials, detection methods, linear range (the range in which the final output value of the system is proportional to the concentration or activity of the analyte), limit of detection (the smallest amount or concentration of analyte in the test sample that can be reliably distinguished from zero), and response time (the time required for the sensor to reach 90% of the final stable value) of these chemical sensors are discussed in detail. According to the selectivity, sensitivity, and stability of the sensors, the existing problems of these sensors and possible solutions are proposed. The organization is as follows: Section 2, Section 3 and Section 4 analyze the research of chemical sensors for the detection of H_2_S, which are classified based on the properties of H_2_S: metal affinity, reducibility, and nucleophilicity. Section 5 discusses methods to improve the performance of chemical sensors and summarizes the future challenges and development prospects of chemical sensors in the field of biological H_2_S detection. This review discusses chemical sensors for detecting H_2_S as a gasotransmitter, which is of great significance for the further innovation and development.

## 2. Metal Affinity Based H_2_S Detection

The metal affinity of H_2_S is mainly manifested in that sulfur ions react with various metal ions (such as Cu^2+^, Ag^+^, Pb^2+^, Zn^2+^, etc.) to produce metal sulfide precipitation. Because of the short reaction time, it is also the most common way to detect H_2_S. Table 1 shows some of the chemical sensors based on metal affinity that can be used to detect H_2_S.

In numerous metal sulfide precipitates, CuS is a very stable substance having a low solubility product constant Ksp = 6.3 × 10^−36^ [18]. In 2015, Meng et al. [19] developed a new fluorescence chemical sensor based on 7-nitrobenzene-2-oxa-1,3-diazole (NBD) fluorophores and introduced Cu^2+^. The presence of Cu^2+^ quenched the fluorescence of fluorophores. When H_2_S was added, the fluorescence intensity recovered at 519 nm due to the generation of CuS. When the concentration of H_2_S was in the range of 0–20 μM, the relationship was linear with the intensity of fluorescence emission at 519 nm. Moreover, it had a relatively low detection limit of 0.17 μM, and the sensor was successfully applied in MDA-MB-231 cells in vivo. In 2016, Guo et al. [20] employed EuW_10_O_36_·32H_2_O (Eu-POM) and 1,2-bis(3-hexadecylimidazolium-1-yl) ethane bromide ([C_16_-2-C_16_im]Br_2_) modified with Cu^2+^ to prepare a supramolecular chemical sensor (CSS). When CuS was formed, the absorbance was increased at 400 nm, accompanied with the conversion of the colorless solution to brown. The linear relationship was between 1.25 μM and 175 μM, and the detection limit was 1.25 μM. In 2018, Wang et al. [21] introduced Cu^2+^ into naphthalimide-rhodamine B and developed a fluorescence colorimetric sensor called 1-Cu^2+^ which was applied to the detection of H_2_S in HeLa cells. When adding H_2_S to produce CuS, on the one hand, the absorbance decreases at 546 nm, the color of the solution changes from pink to light yellow, and a good linear relationship with 0–80 mM H_2_S with a detection limit of 0.4 mM was achieved. On the other hand, the fluorescence intensity at 610 nm was quenched, from bright yellow to green, and had a good linear relationship with 0–40 mM H_2_S, and the detection limit was 0.23 mM. This sensor had a high detection limit and was not suitable for the detection of organisms or cells with low H_2_S content. Detection limits need to be further improved.

In addition to producing copper sulfide precipitates, silver sulfide (black) and lead sulfide (brown) have obvious color changes; it is also one of the options for detecting H_2_S. In 2018, Zhao et al. [22] developed a highly selective and sensitive electrochemical sensor based on Au@Ag core-shell nanoparticles (Au@Ag NPs). The electrical activity of Au@Ag NPs provided good electrical conductivity and promoted the formation of Ag_2_S, resulting in the decrease of differential pulse voltammetry (DPV) peak at 0.26 V. In the range of 0.1 nM to 500 nM, the logarithm of H_2_S concentration was linear with the DPV peak at 0.26 V. In 2019, Jornet-Martínez et al. [23] proposed a method to detect H_2_S by supporting silver nanoparticles (AgNPs) on nylon, which can exhibit good color intensity due to the higher retention rate of the nylon carrier. The formation of Ag_2_S led to further aggregation of monodisperse silver nanoparticles. The absorbance ratio at 550 nm and 415 nm exhibited a linear relationship with the logarithm of H_2_S at 150–1000 ppbv, and the detection limit was 45 ppb. This sensor can detect H_2_S at the ppb level, which had a low detection limit. In 2018, Cha et al. [24] made colorimetric chemical sensors using porous Pb(Ac)_2_ anchored nanofibers (NFs). The schematic diagram of sensor preparation and detection is shown in 1–3 of Figure 3A. It used electrospinning technology to provide a large number of reaction sites for detection. The scanning electron microscope (SEM) image of HTS-Pb(Ac)_2_@NFs synthesized by electrospinning technology is shown in 1 and 2 of Figure 3B. The formation of PbS changed the color, and the color changed under different H_2_S concentrations; exposure times are shown in 1–3 of Figure 3C. H_2_S as low as 400 ppb was detected at a relative humidity of 90%. It was also used to detect H_2_S in breath samples of patients with bad breath.

In addition, in 2013, Wu et al. [25] developed a phosphor chemical sensor using Mn-doped zinc quantum dots. H_2_S was added to quench orange phosphor. The fluorescence intensity at 585 nm was linearly related to H_2_S at 2–100 μM. And the detection limit was 0.3 μM, it was suitable for the detection of H_2_S content in general physiological conditions. In the above tests, the sensor showed good selectivity in anions and small molecules of mercaptan.

The chemical sensor based on metal affinity of H_2_S has the following problems. The following discussion summarizes the previous studies and proposes possible solutions.

### 2.1. Accuracy and Reliability

H_2_S is involved in many physiological and pathological processes, so the accuracy and reliability of sensors need to be improved to make the detection results more convincing and accurate. The ratio detection method plays the role of internal correction, improving the accuracy and reliability of detection. In particular, the fluorescence chemical sensor can effectively reduce background autofluorescence interference and avoid false positives or false negatives [26]. For example, in 2016, Yang et al. [27] developed a ratio fluorescence chemical sensor named HSip-1@AuNCs, using bovine serum albumin–templated gold nanoclusters (BSA-AuNCs) as an internal reference fluorophore and HSip-1 based on aromatic ring Cu^2+^ complex as a H_2_S recognition unit. HSip-1@AuNC emitted dual fluorescence at 519 nm and 632 nm, respectively, coming from HSip-1 and AuNCs. CuS was formed after the addition of H_2_S, which increased the fluorescence intensity at 519 nm but remained unchanged at 632 nm. The linear range was 7–100 μM, the detection limit was 0.73 μM, and it was used for the selective detection of H_2_S in fetal bovine serum. In 2019, Liu et al. [28] designed a nanoratio fluorescent chemical sensor called Ru-Cu@FITC-MSN, which was prepared by immobilizing a luminescent Ru^2+^ complex into a fluorescein isothiocyanate (FITC) conjugated water-dispersible mesoporous silica nanoparticle (MSN), and Cu^2+^ was introduced by the in situ formation method. This sensor showed dual emission bands at 520 nm (FITC) and 600 nm (Ru complex). The addition of H_2_S restored the quenching fluorescence, increased the fluorescence intensity at 600 nm and kept the fluorescence intensity at 520 nm. The linear range was 0.5–4 μM, the detection limit was 0.36 mM, and the limit of quantification was 1.21 mM. The limits of detection and quantification were at the millimolar level, and it was relatively high and was not conducive to the detection of H_2_S at lower levels. It showed good selectivity in anions and sulfur-containing amino acids when detecting H_2_S in breast cancer cells MCF-7. Shang et al. [29] quantitatively detected H_2_S based on the ratio of the oxidation peak current of Ag to MB. MB acted as an internal reference material, providing built-in correction to avoid interference from internal and external environments.

### 2.2. Sensitivity

The sensitivity of the sensor has always been one of the key parameters to evaluate its performance. The content of H_2_S in different parts of the organism is not the same, and sometimes the nanomolar level of H_2_S needs to be detected according to the actual demand [30], which puts forward a higher requirement for the sensitivity of the sensor. The sensitivity can be greatly improved by the detection methods and test materials.

Due to electrochemical (EC) and photoelectrochemical (PEC) signal responses have the advantages of ultra-high sensitivity, low cost, fast detection, etc. In 2021, Xu et al. [31] utilized TiO_2_/Bi_2_WO_6_/Ag heterojunction to prepare a high sensitivity EC-PEC driven dual-mode chemical sensor, the material synthesis is shown in Figure 4A. The sensing process is shown in Figure 4B, and 1–2 in Figure 4B, respectively, show the mechanism of photocurrent signal on–off sensing in the absence and presence of H_2_S. On the one hand, for electrochemical detection, TiO_2_/Bi_2_WO_6_/Ag displayed a highly electrocatalytic activity toward the reduction of H_2_O_2_ through amperometric i–t curve (i–t) due to the nanoenzyme property and synergistic effect, which endowed electrochemical sensing. On the other hand, for photoelectrochemical detection, charge carriers were produced when the light was irradiated onto the TiO_2_/Bi_2_WO_6_/Ag surface electrode. Moreover, the superfast electron transfer is promoted due to the cascade edge level, and the recombination of electron and hole (e−/h+) is effectively inhibited. When H_2_S was added, due to the generation of Ag_2_S and Bi_2_S_3_, the original catalytic activity is reduced; in addition, due to the mismatch between the energy band level and TiO_2_, the recombination of photogenerated electrons and holes is accelerated, finally leading to the sharply decreased double signals. This sensor showed a wide linear response between 0.5–200 μM, a detection limit of 0.08 μM, and a good selectivity for common anions and sulfides.

Electrochemiluminescence (ECL), as an electrochemical technology combined with chemiluminescence detection technology, which has the advantages of high sensitivity and simple instrument. In 2022, Chen et al. [32] introduced Ag^+^ based on cytosine sequence and developed sensors using ECL technology to measure H_2_S in joint fluid. The formation of Ag_2_S triggered hybridization chain reaction, which further produced an electrochemical luminescence signal. The linear response over the concentration range of 0.1000–1500 nM, a detection limit of 0.0398 nM, and a good selectivity in the physiological activity of interferes were obtained. After the sensor was placed for a week, the detection signal decreased by less than 6%, showing good stability. However, the response time was 90 min, necessitating a further improvement.

In addition, sensitivity can also be improved by means of transferring detection targets. In 2015, Gao et al. [33] developed a chemical colorimetric sensor based on the core-shell structure Au@TPt-NCs, which used H_2_O_2_-3,3′,5,5′-tetramethylbenzidine (TMB) product concentration to achieve indirect detection of H_2_S concentration in bovine serum. Au@TPt-NCs is a kind of nanocatalyst, and TMB under the catalytic effect produces the product TMBox. When H_2_S was absorbed by the surface of Pt nanostructure, the surface active site of the catalyst was blocked, the TMBox products were reduced, which further led to the decrease of absorbance at 650 nm, and then the color changed. The detection of 100 nM H_2_S in 10 μL of solution was finally translated to the detection of 70.79 μM TMBox in 100 μL of solution. Thus, the signal was improved about 7079 times. The linear concentration range was 10−100 nM, the slope of the regression equation was 0.013, and the detection limit was 7.5 nM. A good selectivity, reproducibility, and accuracy were also exhibited. 

Mulit-component composite materials can also help improve the sensitivity of sensors due to their excellent properties. In 2022, Li et al. [34] used cellulose nanofiber (CNF)-templated CuO (CuO-C), which was decorated with tungsten disulfide (WS_2_) nanosheets (W-Cu-C) as the sensing layer of electrochemical sensor for H_2_S recognition. A schematic diagram of the material preparation procedure is shown in Figure 5. The combination of metal oxide CuO and transition metal dihalide WS_2_ can significantly sensitize the gas sensor, thus improving the sensitivity of the sensor. On the one hand, CNFs were adopted as the nucleation template of the CuO material to boost the utilization efficiency of the sensing component when interacting with H_2_S molecules simultaneously augmenting the conducting pathways after its calcination into carbon nanofibers. On the other hand, layered WS_2_ nanosheets acted as dual roles of sorption-sites supplier and heterojunction ingredients, jointly amplifying the sensing signal. This sensor had a low detection limit of 200 ppb and an ultra-low power consumption of 8 mW, which provided a good choice for wearable devices. 

In 2020, Shang et al. [29] used electroactive Ag dendriticnano composites (NCs) as a special recognition element of H_2_S with methylene blue (MB) as an inner reference molecular, and then combined with graphene oxide (GO) to develop an electrochemical sensor. Dendritic nanostructures have a larger surface area and more active sites while GO possesses a good adsorption capacity, thus improving the sensor sensitivity. Electroactive Ag NCs modified the GCE to act as a selective recognition probe of H_2_S. Dendritic Ag NCs reacted with H_2_S to form Ag_2_S, then reduced the electrochemical oxidation peak current of Ag (0.42 V) and kept the current response of MB constant. GO/MB composite was optimized as an inner reference element to provide a built-in correction for avoiding biomolecular interference. The ratio of Ag and MB oxidation peak current of H_2_S at 30–500 μM was linear, and the linear range was higher than that of the conventional electrochemical method. The detection limit was 23.88 μM, which was relatively high compared with other sensors that can detect a few tenths of a micromole or even a nanomole of H_2_S. This sensor showed good selectivity and stability within the detection of H_2_S in bovine serum.

The application of different detection techniques improves the sensitivity but also puts forward higher requirements for the professionalism of detection personnel. The price of detection materials with excellent performance increases the cost of sensors. Therefore, new methods need to be developed to improve the sensitivity of detecting H_2_S.

### 2.3. Biocompatibility

Due to the presence of metals or metal ions in the test material, issues of biocompatibility and toxicity must be considered. Small amounts of metal ions are not toxic. However, at high concentrations of H_2_S, more metal ions are needed to produce further metal sulfides. The presence of too many metal ions can be toxic to organisms or lead to a decrease in cell activity [35].

In order to solve the problem of biocompatibility and toxicity, it can be solved from two aspects. The first is composites engineering with secondary substances of good biocompatibility and low toxicity. In 2016, Ding et al. [36] developed an electrochemical sensor based on TiO_2_ to detect endogenous H_2_S production in stimulating MCF-7 cells with VEGF. The synthesis and detection process of the sensor is shown in the Figure 6A. The surface of TiO_2_ nanotubes was modified by a layer of thioglycolic acid (TGA) to introduce thiol groups which can adsorb and immobilize Cd ions. Because of the good biocompatibility, TiO_2_ greatly improves the cell activity when detecting exogenous H_2_S released by cancer cells. However, the large band gap (3.0–3.2 eV) of TiO_2_ limits its adsorption mainly in UV range. Figure 6B clearly elucidates the injection of excited electrons from CdS to TiO_2_. When CdS is generated, it also acts as a visible light sensitizer due to its narrow band gap (2.4 eV), which improves the sensitivity. The SEM images, TiO_2_ nanotubes, and those treated after with different concentrations of H_2_S, are shown in 1–3 of Figure 6C. 

The sensor has good selectivity and achieves a wide linear range of 10–106 nM with a detection limit of 0.7 nM. The detection limit of nanomole level can realize the accurate detection of H_2_S at the physiological level of micromole level. In 2020, Hao et al. [37] reported the design of a peptide L (FITC-AhxHis-Glu-Phe-His-NH2) and introduced Cu^2+^ to design a fluorescent chemical sensor based on L: Cu^2+^ with good selectivity for H_2_S detection. Peptide-based chemical sensors are more biocompatible, and they are generally less toxic and exhibit good cellular permeability. High concentration of L: Cu^2+^ was used to detect H_2_S in HeLa cells, and the cell viability remained above 95%. After the addition of H_2_S to produce CuS, the green fluorescence recovered, while a linear relationship between the fluorescence intensity at 520 nm and 0–0.7 μM H_2_S was achieved, with a detection limit of 68 nM.

The second aspect is to employ metals with good biocompatibility. For example, metal gold, especially gold nanoclusters (Au NCs), has low toxicity and good water solubility and biocompatibility. In 2015, Liang et al. [38] prepared a dual-emission chemical sensor by covalently attaching fluorescent carbon nanoparticles (CNPs) to Au NCs, triggering the sensing mechanism of fluorescence resonance energy transfer (FRET) from CNPs (donor) to Au NCs (acceptor). Ideal reaction results were demonstrated at PH = 7.4 upon the detection of H_2_S in serum samples. Additionally, good selectivity in anions and amino acids was presented. The formation of Au_2_S increased the fluorescence intensity at 445 nm and decreased at 575 nm. The detection limit was 18 nM when H_2_S of 0–60 μM was linearly related to the ratio of the fluorescence intensity at 445 nm and 575 nm. In 2017, Zhang et al. [39] designed a fluorescent chemical sensor based on acetylcysteine-stabilized gold nanoclusters (ACC@AuNCs) to detect H_2_S in serum samples. Gold nanoclusters (NCs), which can produce unique optical properties and have some special properties such as large Stokes shift, strong fluorescence, high photochemical stability and wide spectrum lines, have been the preferred choice for H_2_S nanosensors. Figure 7A illustrates the detection process of the fluorescent sensor. This sensor showed higher H_2_S selectivity as compared to that toward other anions, amino acids, and mercaptans, as shown in Figure 7B. The sensor presented a stable response over the concentration range from 0.002 to 120 μmol/L (Figure 7C) with the detection limit of 1.8 nM/L. Nevertheless, a long detection time remained.

In addition, metal bismuth has non-toxic properties. Bi^3+^ has been used as a reagent in Pepto-Bismol and other drugs. In 2016, Anal et al. [40] used Bi(OH)_3_ and its derivatives to capture H_2_S in breath samples, and the production of Bi_2_S_3_ changed the color from white to yellow/brown. This sensor responds to ≥30 ppb H_2_S in a total volume of 1.35 L of gas, and it is at least two orders of magnitude more sensitive than a commercial H_2_S test paper based on Pb^2+^(acetate)_2_, but this sensor can also detect sulfide and mercaptan, thus lacking good selectivity.

Although the above methods provide good biocompatibility for H_2_S detection to a certain extent, the sensitive material system increases the complexity and time of preparation. Moreover, metals with good biocompatibility, especially metal nanoparticles, have the disadvantage of high price. Therefore, detecting H_2_S with metal affinity still needs to be further solved.

### 2.4. Detection Medium

Many H_2_S chemical sensors, especially fluorescence or colorimetric chemical sensors, usually realize the H_2_S detection in a liquid medium. If the detection medium can be transferred from a liquid phase to a solid one, the complexity of detection is greatly simplified, which provides an easy access to non-professionals. At the same time, fluorescence and colorimetric sensing can offer more obvious and clear visual results more suitable for wearable devices. The sensor is made of thin film with a paper base and a H_2_S-specific detection reagent. The solid detection platform provides a new road for wearable devices.

In 2016, Chen et al. [41] used polystyrene sulfonate (PSS), a strong polyelectrolyte, to prepare red-photoluminescent PSS-penicillamine (PA) copper (Cu) nanocluster (NC) aggregates, which displayed high selectivity and sensitivity toward H_2_S detection. During the design process, the paper-based microfluidic analysis device was utilized to transfer the detection platform from the solution to the paper base. The device was inexpensive, portable, and requires only a small number of samples (5 μL). In 2019, Sarfraz et al. [42] designed an inexpensive disposable H_2_S sensor on paper substrate with the diagram shown in the Figure 8.

This sensor was produced by inkjet printing copper-acetate-based sensing films on ultra-thin gold film electrodes (UTGFE) with a thickness of 20 nm and gold electrodes (GE) with a thickness of 40 nm. Inkjet printing technology improves the electrode coverage and makes the reaction better. With the increase of H_2_S concentration, the color of the paper base changed visible to the naked eye (from blue green to gray brown). As low as 1.5 ppm H_2_S can be detected within 5 min, and the linear range was 0–200 ppm.

In 2022, Ferrer et al. [43] improved the detection device of AgNPs retained on a Nylon surface sensor designed by Jornet-Martinez et al. [23]. A pair of sensors and a magnet were introduced on a 5 × 7 cm bag and applied to the detection of H_2_S in breath gas, showing a color change from yellow to brown, with a detection limit of 0.019 mg/L and the limit of quantitation of 0.06 mg/L, but when humidity drops below 20%, the sensitivity of the sensor decreased.

Moving the liquid-phase detection medium into the solid one opens up more possibilities for wearables, but most H_2_S detection is currently performed in liquids where there is less interference. Therefore, when the liquid phase detection is transferred to the solid phase, it is necessary to prevent other gases from affecting the detection, which puts forward higher requirements for the selectivity of the sensor.

### 2.5. Repeatability

At present, one of the major difficulties in H_2_S detection is the sensor repeatability. Many sensors are disposable and need to be reprepared before each detection, which cannot be reused, increases the production cost and time to a certain extent. Many researchers have also come up with solutions to this problem.

In 2016, Li et al. [44] prepared porous CuO nanosheets on alumina tubes by a facile hydrothermal method to produce an electrochemical sensor. The 2D sheet-like CuO nanostructures have high anisotropy and nanoscale thickness. The nanoporous structures of CuO sheets could allow a fast and efficient gas adsorption on their surfaces, so the process of adsorption and desorption can be repeated in a short time. This sensor was continuously exposed to 200 ppb H_2_S 5 times at room temperature, and showed good repeatability after continuous adsorption and desorption process. In 2017, Xu et al. [45] designed and synthesized a novel fluorescent chemical sensor HACBA-Cu^2+^ with carbazole-hemicyanine fluorophore as signal reporter and *N*,*N*,*N*’-tri(2-pyridylmethyl)ethylenediamin e (TPEA) as binding site. Alternately, added H_2_S and Cu^2+^ can achieve an “on-off-on” form of fluorescence cycle, and at least 3 repeat cycles were observed, but the fluorescence intensity decreased after multiple repeats. In the same year, Strianese et al. [46] developed a fluorescent sensor using the zinc porphyrin complex TMPyPZn, which promoted the reversibility of H_2_S binding with zinc phthalate by adding acetic acid. Although the sensor could be repeated many times, the fluorescence intensity level slowly decreased. In 2018, Dulac et al. [47] developed a fluorescence sensor using the hemoglobin I (HbI) for the clam to detect H_2_S in plasma. Due to the reversible coordination between H_2_S and HbI, the addition of argon promotes the reversible reaction and restore the fluorescence.

Although the above methods endowed the sensor with a certain repeatability, most of them realize repeated measurement only in a short time and cannot be detected after long-term storage. In addition, the fluorescence intensity decreases after repeated detection, which makes the performance of the sensor deteriorate. Finally, in order to restore the sensors to their original state, additional chemicals are added, probably bringing toxicity. Therefore, the repeatability of such sensors is still a big problem to be solved in the future.

## 3. Reducibility Based H_2_S Detection

H_2_S chemical sensors based on reducibility can be divided into two categories: one is to reduce azide (-N_3_) groups to amino (-NH_2_) under the action of H_2_S [48,49,50,51] (belongs to reduction-oxidation reaction), commonly used in fluorescent sensors; the other is the reduction-oxidation (REDOX) reaction of H_2_S with other oxidizing substances [52,53,54], which is often used in electrochemical sensors. Table 2 shows some of the chemical sensors based on reducibility that can be used to detect H_2_S.

As for, fluorescence sensors, fluorescent dyes containing azide groups are often used as detection materials, which are reduced to amino group under the action of H_2_S. The originally quenched-fluorophore can restore fluorescence and achieve the detection effect visible to the naked eye. For example, in 2013, Sun et al. [55] used 2-allyl-1,3-dioxo-2,3-dihydro-1h-Benzo[de]isoquinoline 6-sulfonyl azide (AISA) copolymerized with styrene, a fluorescent chemical sensor named PSAISA was designed. In the presence of H_2_S, sulfonyl azide was reduced to sulfonamide, which enhanced the intensity of fluorescence emission at 530 nm, resulting in the color from colorless to bright yellow. When applied to the detection of H_2_S in HeLa cells, and good selectivity was achieved in anions and sulfur-containing amino acids. In 2016, Kim et al. [56] designed a fluorescent sensor called AHS based on 2,3-naphthalimide (2,3-NI) to selectively detect H_2_S in histiocytic lymphoma cells U937. The azide in 2,3-NI was reduced to amino group by H_2_S, resulting in increased fluorescence intensity at 557 nm and the linear range was 0–80 μM. In 2019, Zhang et al. [57]. coupled the spiropyran derivative to the weakly fluorescent molecule 4-azide-1,8-naphthalic anhydride with ethylenediamine as a tether to synthesize NT-N_3_-SP for specific monitoring of H_2_S. The addition of H_2_S reduced the azide group, the maximal absorption at 370 nm decreased and a new red-shifted absorbance peak at 440 nm was centered, fluorescence intensity increased significantly at 540 nm. Stable detection of H_2_S can be performed at PH = 3–8, and it has been successfully used to screen H_2_S in living mice. 

In 2022, Jothi et al. [58] used 6-azido-2-(pyridin-2-ylmethyl)-1h-Benzo[de]isoquinoline-1,3(2H)-Dione (NPN-N_3_) for fluorescence detection of H_2_S in Escherichia coli cells. A and B in Figure 9, respectively, show the synthesis of NPN-N_3_ and the emission response mechanism of NPN-N_3_ to H_2_S. The azide group of nonfluorescent napthalimide was reduced to amino group under the action of H_2_S, and the green fluorescence was produced. In Figure 9C, 1–3 shows the change of fluorescence spectrum and the linear relationship after the addition of H_2_S or different anions, and 4 shows the change of solution color. In addition, the absorbance of the sensor probe decreased at 365 nm and redshifted to 416 nm after H_2_S exposure, accompanied with naked eye-visible color change from colorless to yellow. This phenomenon indicated that the probe could also be used as a colorimetric sensor.

In addition, due to the reducibility of H_2_S, REDOX reactions often occur with functional groups or substances with oxidation properties in electrochemical detection. The oxidation products of H_2_S lead to changing the electrode resistance, and H_2_S detection is realized through signal processing. For example, in 2018, Jackson et al. [59] used the triple pulse amperometry (TPA) method to electro-oxidize H_2_S to SO_4_^2−^ at glass carbon electrodes for the detection of potential H_2_S in biological systems. In 2022, Kim et al. [60] contained carbon-nanofiber (Fe_2_O_3_-MPCNF)-based conductive paste (the preparation process is shown in Figure 10A) through an iron oxide-immobilized multiscale pore, and the addition of H_2_S reacted with the adsorption oxygen functional groups in Fe_2_O_3_ to form SO_2_; the reaction and recovery mechanisms are shown in the Figure 10B. The REDOX reaction produced electrons and entered the carbon structure, causing resistance to increase. The sensor could detect H_2_S of 0.2–100 ppm at room temperature. This sensor was also combined with the wireless system to realize the wearable wireless H_2_S detection.

Concerning the reducibility of H_2_S, the following content summarizes the previous studies and proposes possible solutions.

### 3.1. Accuracy and Reliability

Due to the complex biological environment, the detection results may be affected by the fluorescence interference when the fluorescence sensors are applied.

Ratio detection can effectively eliminate the background interference and the fluctuation of detection conditions caused by the sample’s own fluorescence or instrument factors and play a role of self-calibration [61] so as to obtain more reliable detection. In 2012, Yu et al. [62] proposed ratio fluorescence sensors based on the structure of CD-naphthalimide (CDs)-azide for selective H_2_S detection HeLa cells (human cervical cancer cell) and L929 cells (murine aneuploid fibro sarcoma cell). After the azide group was reduced by H_2_S, the emission intensity ratio of 526 nm to 425 nm increased with the increase of H_2_S concentration. The fluorescence changed from blue to bright green within 15 min and had a low detection limit of 10 nM, but the sensor did not show a good linearity. In 2019, Youssef et al. [63] added p-azide phenylalanine (pAzF) to fluorescent protein (FPs) to design a ratio fluorescence sensor that showed a good ratio response in mammalian cells. There was a linear relationship between 1–100 μM H_2_S and the emission intensity ratio of 450 nm to 500 nm. However, this sensor had some limitations, such as decreased cell activity after detection and the incompatible fluorescent dye with cells or cell lines that were difficult to transfect.

A series of methods have also been proposed to solve the problem of interference, when azide is reduced to amino by H_2_S. In 2016, Yao et al. [64] used time-gated detection in combination with ligand-sensitized complexes of lanthanide cations to eliminate fluorescent interference. In addition, using near-infrared fluorescent dyes to detect cell autofluorescence is not an issue because of low background fluorescence interference. The light used for excitation causes much less photodamage to cells compared to ultraviolet or visible light used in excitation of other dyes. Additionally, a higher signal-to-noise ratio and better tissue penetration were implemented [65]. In 2014, Ozdemir et al. [66] developed a fluorescence sensor based on near-infrared Bodipy dye to detect H_2_S in MCF-7 cells and FBS cells. Bodipy dyes, which are difluoroboron-cHeLated dipyrromethene derivatives, to be good choices for H_2_S detection due to their desirable properties, such as high quantum yield, chemical and photochemical stability, high molar absorption coefficient, and they allow direct access to near infrared emission derivatives. Bodipy 5 was obtained by the reaction of aldehyde 4 and 3-ethyl-2,4-dimethylpyrrole, when H_2_S was added, the azide was reduced to an amine, resulting in fluorescence quenching and a 20 nm redshift. It showed good selectivity in common anions and reactive oxygen/nitrogen species, and the detection limit was 0.34 μM. In 2018, Ji et al. [67] designed a fluorescence sensor based on nimazide to detect H_2_S in mitochondria. Nimazide was synthesized by introducing sulfonyl azide to the core structure of a QSY-21 dark quencher. On the one hand, nimazide as a near-infrared fluorescent dye can reduce phototoxicity and increase tissue penetration. On the other hand, when its azide functional group is connected with electron-absorbing groups such as sulfonyl groups, its reaction activity to H_2_S is greatly enhanced, and the reaction speed is three to four orders of magnitude faster than that between aryl azide and H_2_S. H_2_S could reduce nimazide into a sulfonamide, which would further cyclize to form a spiral lactam, thereby leading to the decrease of near-infrared fluorescence. The sensor had a wide linear range of 10 nm–10 μM, showing good selectivity among common competitive analytes.

Near-infrared fluorescent dyes can reduce the interference of external fluorescence to a certain extent, but they have the following disadvantages: First of all, the toxicity of dyes needs to be considered, and the number of available dyes is even smaller. Secondly, some near infrared fluorescent dyes have poor chemical and light stability, such as cyanine dyes. Finally, the solubility of many near-infrared fluorescent dyes such as borodipyrrole dyes is difficult to meet the requirements of practical detection [68]. If we want to use near-infrared fluorescent dyes to detect H_2_S, more research needs to be conducted on dye synthesis.

### 3.2. Sensitivity

In electrochemical H_2_S sensors, if the detection material has a larger surface area or specific surface area, it provides more active sites. At the same time, the electrode material has a high conductivity, and the electrode current changes quickly, so that the sensitivity of the electrochemical sensor can be improved to a large extent.

In 2019, Asif et al. [69] designed an extremely sensitive electrochemical sensor with a 2D nanosheet-shaped layered double hydroxide (LDH) wrapped carbon nanotubes (CNTs) nanohybrid (CNTs@LDH), where a series of CNTs@CuMn-LDH nanohybrids with varied amounts of LDH nanosheets grafted on a conductive CNTs backbone had been synthesized via a facile coprecipitation approach. The material synthesis and testing process is shown in the Figure 11A. Nanomaterial non-enzyme electrochemical sensors show excellent performance in sensitivity and anti-interference ability. CNTs@CuMn-LDH had superior surface area (different electron microscope images are shown in 1–4 of Figure 11B), providing more binding active sites for H_2_S. Meanwhile, the high conductivity of carbon nanotubes improved the efficiency of the oxidation of H_2_S into S, further greatly improving the sensitivity. This sensor can detect at low potential, reduce the interference of electroactive substances in the organism to detect, and was applied to the H_2_S secreted by human cells stimulated by VEGF. The linear range was 8 nM–2.9 mM, the detection limit was 0.3 nM. It had wide detection range and low detection limit.

In the same year, Jha et al. [70] reported liquid-exfoliated MoSe_2_ nanoflakes based stable electrochemical sensor for the detection of H_2_S in human exhaled gas. The exfoliated MoSe_2_ nanosheets exhibit P-type behavior and adsorbed oxygen can change the initial conductivity of the sensing material, and thus the sensor had more adsorption sites. Being a reducing agent, H_2_S donates electron to the MoSe_2_ nanoflakes, effectively counterdoping, which decreases the current. The detection limit was 6.73 ppb, the limit of quantification was 22.44 ppb, and the sensor sensitivity was 5.57%/ppm, catering for a near-real-time detection. This was a big improvement over the previous H_2_S gas sensor. In 2021, Liu et al. [71] successfully synthesized multilayer Ti_3_C_2_T_x_ MXene graphene-like structure by simple hydrogen fluoride etching method and fabricated it on glass carbon electrode (GCE) for electrochemical detection of H_2_S in biological environment. The layered structure of the two-dimensional transition metal carbon/nitide has the advantages of good conductivity and large specific surface area, providing more reaction sites. H_2_S is oxidized to S on the electrode surface, with a high sensitivity of 0.587 μA μm^−1^ cm^−2^ and a wide linear detection range of 100 nM–300 μM. This sensor can detect H_2_S as low as the nanomolar level.

### 3.3. Response Time

The essence of REDOX reaction is electron gain, loss, or migration. H_2_S, as a reducing substance, loses electrons in REDOX reaction. Therefore, if the detection material can be connected with the electron-withdrawing group, the electron loss process of H_2_S is accelerated to a certain extent, then the response time of the sensor is shortened. The short response time benefits a real-time detection and wearable detection systems.

In 2016, Choi et al. [72] introduced an electron-withdrawing group, fluorine, on the o-position of the ‘N-imide site’ in order to improve the reaction rate. The response time was two times better than that of the sensor without the electron-absorbing group. The sensor was used to selectively detect H_2_S in macrophage RAW264.7 with a linear range of 0–200 μM and a detection limit less than 0.3 μM. In 2018, Ji et al. [67] connected azide functional groups to sulfonyl electron-withdrawing groups, leading to an accelerated response speed toward H_2_S. The reaction rate was three to four orders of magnitude faster than the reaction rate between aryl azide and H_2_S, and it can detect H_2_S at nanomolar levels. In addition to introducing electron- withdrawing groups, efficient catalytic systems can also reduce response times. In 2018, Kim et al. [73] designed a H_2_S electrochemical sensor based on semiconductor metal oxides (SMOs), they fabricated superior H_2_S sensing layers using dual sacrificial biotemplates, i.e., cellulose nanocrystals and apoferritin, in a one-pot synthesis of cocatalysts (PtNa_2_W_4_O_13_) functionalized WO_3_ nanotubes using electrospinning technique. The dual-catalytic system of Na_2_W_4_O_13_ and bio-inspired Pt catalyst accelerated the reaction of H_2_S with chemisorbed oxygen, with a response time of 5–10 s, which was improved by 2–6 times as compared with the traditional SMOs sensor. Sensing layers based on electrospun nanofibers can offer high gas accessibility and large surface area, resulting in active interaction of air adsorbates with gas analytes, and it can also shorten the response time to some extent. The detection limit was 3 ppb, which was suitable for H_2_S detection in the exhaled breath.

### 3.4. Water Solubility

As is well known, water is the most abundant compound in cells. To increase the water solubility of the corresponding probe in the fluorescent sensor is conducive to real-time in-vivo H_2_S detection and imaging.

In 2015, Xu et al. [74] introduced a hydrophilic alcoholic group to the naphthalimide core to improve water solubility and showed good cell membrane permeability. This sensor was used for selective detection of H_2_S in HeLa cells at the cost of long response time of 15 min. In 2018, Zhang et al. [75] designed an Al-MIL-53-NO_2_ fluorescence sensor from metal-organic framework (MOF)-polymer mixed-matrix membranes (MMMs). The detection materials were evenly distributed within the pores of MOF. Due to the high porosity, the MMMs have good water permeability flux so that H_2_S in the body can be completely contacted with MOF. In 2019, Ma et al. [76]. used quinoline as fluorophore to improve the water solubility in H_2_S detection in view of its excellent optical properties, good solubility, and low cytotoxicity. With the addition of H_2_S, the fluorescence intensity was enhanced 11 times at 533 nm. H_2_S imaging in HepG-2 cells was successfully applied, and 87.7% of cells remained active after 10 h treatment.

## 4. Nucleophilicity Based H_2_S Detection

H_2_S has strong nucleophilic properties and can attack electrophilic atoms. Therefore, H_2_S is often used as a nucleophilic reagent to undergo nucleophilic reactions with test materials (such as nucleophilic addition reaction, nucleophilic substitution reaction or thiolysis reaction) [77,78,79,80,81] so as to realize the H_2_S detection. Table 3 shows some of the chemical sensors based on nucleophilicity that can be used to detect H_2_S.

In 2014, Yang et al. [82] designed a H_2_S selective colorimetric and fluorescence sensor based on diethylaminocoumarin–hemicyanine dye. The addition of H_2_S carries out nucleophilic attack on polarized C=N and electron-poor C=C in the dye, then made the UV-vis absorption of the sensor taking place blue-shift from red to yellow, and the fluorescence of the sensor also changed from red to green. The PH value of 7–11 was the effective detection range when detecting H_2_S content in HepG2 cells with the low detection limit of 14 nM. This was the first report on the nucleophilic H_2_S added to the polarized C=N bond and electron-poor C=C bond of the hybrid coumarin–hemicyanine dye.

In 2015, Karakuş et al. [83] designed an ESIPT based fluorescent dye, 3-hydroxyflavone, that was chemically masked by an electrophilic cyanate motif. The cyanate (O-CN) unit acted as the specific recognition motif of H_2_S. When H_2_S was added, nucleophilic addition reaction occurred, resulting in the cleavage of carbon and nitrogen triple bond in the masking group, at the same time generated the free hydroxyl group, which promoted the generation of fluorescence. It was stable in PH 4–9 and had a detection limit of 0.25 μM. It was used to detect H_2_S produced by human lung adenocarcinoma cell A549.

In 2018, Chen et al. [84] developed a chemical sensor containing fluorobenzene group based on coumarin derivatives that can be used for fluorescence and colorimetric detection of H_2_S. The addition of H_2_S thiolyses with dinitrophenyl ether in the test material. On the one hand, the absorbance decreases at 320 nm while new absorption occurs at 470 nm, together with the color change from colorless to bright yellow, which can be used for semi-quantitative colorimetric H_2_S detection. On the other hand, the emission intensity increased at 392 nm after thiolysis and showed a linear relationship with H_2_S in the range of 0–8 × 10^−6^ M/L, which could be used for quantitative fluorescence H_2_S detection. The sensor was used to detect H_2_S in human breast cancer cells MCF-7 and showed good selectivity and low cytotoxicity.

In 2020, Wang et al. [85] synthesized a novel fluorescent molecule called (E)-6-(4-(diphenylamino)benzylidene)-5-oxo-5,6,7,8-tetrahydronaphthalen-2-yl 2,4-dinitrobenzenesulfonate (BOTD), which was used to design a H_2_S fluorescence chemical sensor. The BOTD synthesis process is shown in the Figure 12A. In BOTD fluorescent molecule, triphenylamine as a strong electron donor and 6-hydroxy-3,4-dihydronaphthalen-1 (2H)-one conjugated to form a fluorescent probe precursor (BOT) and 2,4-dinitrobenzenesulfonyl (DNS) as a response site. When H_2_S is added, DNS undergoes nucleophilic reaction with H_2_S, resulting in the formation of hydroxyl at the end. Electron recovery, obvious yellow–green fluorescence, can be observed by naked eye within 30 s. The reaction process is shown in the Figure 12B. The sensor showed good fluorescence characteristics at PH 6–8, which was suitable for physiological detection. The sensor showed a linear range over 0–50 μM/L, a low detection limit of 27.3 nM/L, and good selectivity in common anions and sulfur-containing amino acids. It was used to detect endogenous and exogenous H_2_S. BOTD was exploited for imaging in HeLa cells, which is shown in the Figure 12C. After 24 h incubation, the survival rate of cells was 85%, revealing a good biocompatibility.

The following discussion summarizes the previous studies and proposes possible solutions regarding chemical sensor based on nucleophilicity of H_2_S.

### 4.1. Accuracy and Reliability

Chemical H_2_S sensors based on nucleophilic reactions also need to solve the problem of detection reliability. In H_2_S fluorescence chemical sensor, it is mainly necessary to eliminate the interference caused by the fluorescence. The most commonly used method is to use ratio detection that concurrently plays a role of self-calibration.

In 2019, Mao et al. [86]. designed a ratiometric and near-infrared fluorescence detection method based on isophorone and was successfully applied in cell imaging. After the addition of H_2_S, the nucleophilic attack was carried out, which showed the emission ratios of fluorescence intensity at 540 and 660 nm (I540/I660) with an obvious enhancement from 0.05 to 13.3, and the color changed from pink to yellow. The self-calibration function of ratio detection and the low interference of near-infrared dye improved the accuracy and reliability. In 2021, Fan et al. [87] constructed a ratio fluorescence H_2_S sensor with surfactant structure based on fluorescence resonance energy transfer (FRET). Dansyl-based derivative (DBr) is the FRET donor, pyronine Y (PyY) is the FRET receptor and H_2_S reaction site and particularly adopted cationic hexadecyltrimethylammonium bromide (CTAB) micelles to encapsulate and modulate the fluorophores. H_2_S addition completely disrupted the weak FRET process from CTAB-encapsulated DBr to PyY, as shown in Figure 13A, leading to the disappearance of PyY fluorescence emission and the recovery of DBr emission. Meanwhile, the fluorescence intensity increased at 528 nm and gradually decreased and disappeared at 569 nm after H_2_S addition. Under the irradiation of a 365 nm UV lamp, the color of the solution changed from orange to blue–green. The fluorescence intensity and color changes are shown in the Figure 13B. The fluorescence intensity ratio of 528 nm and 569 nm was linearly related to the concentration of H_2_S, and the linear range was 0–15 μM and 20–100 μM. The detection limit was 110 nm, and it was not interfered with by anions and biological mercaptan. The recovery rate of H_2_S in human serum was 98.9–102% by ratio detection.

In addition to ratio detection, multi-channel detection is also one of the methods to improve the sensor reliability. In 2013, Wang et al. [88] developed the first organic three-channel based H_2_S fluorescence sensor. An excited state intramolecular proton transfer (ESIPT) dye was first synthesized from two subunits, 1H-phenanthro[9,10-d]imidazol and 3-hydroxychromone moieties, which were connected by a rigid and conjugated phenyl linker. In general, a single ESIPT dye can provide two different emission channels. The dye synthesized in this work provided emission fluorescence of red and green channels. Then the blue fluorescent dye dinitrobenzene was coupled with the ESIPT dye to synthesize a three-channel organic white-light-emitting dyes. Due to the fluorescence quenching effect of the nitro group, the newly synthesized organic white light-emitting dye is almost non-fluorescent in the three channels. The addition of H_2_S induced a thiolysis reaction with dinitrobenzene, thus enhancing the fluorescence of the red, green, and blue channels by 20, 10, and 3 times respectively. The fluorescence signals of three different channels can be used to verify each other, which can eliminate potential errors better than two-channel detection. The sensor was used for selective detection of H_2_S in human osteosarcoma cells MG63 and showed no cytotoxicity when the sensor dye concentration was less than 50 mM. However, the synthesis process of the dye was complicated, and white light may cause photodamage.

In 2015, Zhang et al. [89] designed the first dual-signal molecular H_2_S sensor using tetraphenylethylene (TPE) as the emission fluorescence source and dinitrobenzene ether as the H_2_S recognition site, which can provide several detection modes of fluorescence, UV-visible absorption, and vision. H_2_S reacts with dinitrobenzene ether, forming hydroxyl group and turning on the fluorescence. On the one hand, the absorbance formed a new absorption band at 450 nm, which changed from colorless to brown visually. On the other hand, the fluorescence intensity increased at 480 nm, showing green fluorescence. The results of ultraviolet absorption and fluorescence signal improved the reliability of detection. The linear range of the sensor was 0.1 μM/L–0.8 mm/L, and the detection limit was 12.8 nM/L. It was applied to detect H_2_S in HeLa cells, and the cells remained 95% active after 12 h. According to the report, the method has been frequently used in H_2_S detection since then, improving reliability while allowing the results to be visualized.

Although the above detection methods are different, their basic idea is basically the same, that is to let several signals mutually confirm the accuracy and reliability of the changes, so as to make the results more convincing.

In addition to the method of using several signals to support each other, in 2018, Ma et al. [90] designed an electrochemical luminescence (ECL) sensor. The sensor did not change the intensity of ECL but used the wavelength of maximum emission (*λ*max) shift as a reading to achieve the quantitative detection of H_2_S. For a given blob, *λ*max in the ECL spectrum is the unique and intrinsic feature. Therefore, the use of *λ*max as a reading can effectively avoid the interference of total luminous intensity fluctuations, providing highly reliable quantitative ECL analysis. In this sensor, the ECL luminophore was fabricated by conjugating Ru(bpy)_3_^2+^-doped silica to oxidized graphene (RuSiO_2_@GO), and the inner filter absorber was a H_2_S-responsive chemodosimeter (CouMC). The reaction between H_2_S and the indolenium C-2 atom of CouMC reduced the absorption of CouMC, while the spectral overlap of the absorption of CouMC and the ECL emission of RuSiO_2_@GO decreased. The suppressed IFE then induced a spectral blue-shift of the ECL emission of RuSiO_2_@GO. The H_2_S sensing process is shown in the Figure 14. In the detection of serum H_2_S, the recovery rate of ECL method based on spectral shift was 99.2%, and that of the strength method was 47.9%. Compared with that of ECL method based on spectral shift, the detection reliability was greatly improved. However, this detection method requires additional equipment (such as CCD) to assist detection, which complicated the equipment.

The application of detection materials without fluorescent background interference can also improve the reliability of fluorescence chemical sensor detection. In 2015, Peng et al. [91] designed a fluorescence sensor based on lanthanide-doped upconversion nanoparticles (UCNPs) binding to luminescent clusters for selective detection of H_2_S in serum. The addition of H_2_S attacks the nucleophilic luminescence group, causing its conjugated structure to be destroyed, leading to fluorescence recovery. UCNPs are often used in biological detection because they have many excellent properties, such as large anti-Stokes shift up to hundreds of nanometers, high light stability, high penetration depth, and most importantly, no automatic fluorescence of biological samples.

In 2017, Yu et al. [92] designed a H_2_S fluorescence sensor named Cy2-MSNs/Ir1 and optimized the detection methods and materials to improve the sensor reliability. Compared with the intensity-based method, the lifetime-based detection is independent of the probe concentration and can efficiently distinguish the signals of the probe from the autofluorescence in complex biological samples. In addition, iridium Ir1 in the detection material is often used as phosphorescent transition-metal complexes (PTMCs) in the background sensing of complex biological samples. The addition of H_2_S produces nucleophilic addition reaction with Cy2, which quenches the fluorescence. The sensor was applied to the detection and imaging of H_2_S in HepG2 hepatoma cells, and the cell activity reached 82% after 24 h, with a low detection limit of 2.7 μM.

### 4.2. Nucleophilic Reaction Time

In 2019, Huang et al. [93] prepared a new amine compound NBD (7-nitro-1,2,3-benzoxadiazole)-amine as a co-reactant, and synthesized Ru(bpy)_3_^2+^ doped silica nanoparticles (RuSi NPs) by water-oil microemulsion method. 

A H_2_S electrochemiluminescence (ECL) sensor was designed by immobilizing RuSi NPs on the surface of a glass carbon electrode using Nafion. After the addition of H_2_S, the thiolysis reaction occurs with the co-reactant NBD-amine to produce the thiolysis product NBD-SH-H, which released diamine. Diamine, as a co-reactant, participates in the electron transfer process of amine to ruthenium on the electrode surface, improving the ECL efficiency (RuSi NPs as a luminescent group). The increase in ECL efficiency improved sensor sensitivity. The manufacturing and reaction process of the sensor is shown in 1 and 2 of Figure 15A. The solution color changed from light orange to rose red as the absorbance decreased at 496 nm and a new absorption peak appeared at 555 nm, which is shown in the Figure 15B. The increase of ECL intensity (I) was linearly related to the logarithm of H_2_S concentration. The linear range was 1 × 10^−10^–10^−5^ mol/L, and the detection limit was 1.7 × 10^−12^ mol/L. It showed good selectivity in the detection of human serum H_2_S, and the recovery rate reached 100–105%. However, after H_2_S was added to the sensor, the ECL signal tended to stabilize for 180 min. The reaction time was too long. The long nucleophilic reaction time brings obstacles to real-time detection, which is one of the problems that some H_2_S sensors need to solve.

Some solutions to the problem of long nucleophilic reaction time are proposed. In 2017, Reja et al. [94] synthesized a molecule with dual nucleophilic center as a fluorescent sensor to increase the speed and reduce reaction time of nucleophilic reaction. In the presence of DIPEA and triphosgene, 2-bromoethanol reacted with methyl-protected fluorescein to obtain dual nucleophilic center molecules. The bromine group acts as the leaving group. When H_2_S was added, it replaced the bromine group by intermediate SH. Then the second nucleophilic substitution reaction operated, in which the thiol group of the intermediate acted as the nucleophilic reagent to eliminate the thiolactone and made the fluorescein ring open to produce fluorescence. The fluorescence intensity tended to be stable at 50 min, which reduced the nucleophilic reaction time to a certain extent. The sensor was used to detect H_2_S in C6 glioblastoma and BV-2 microglia cells, and was successfully applied to imaging H_2_S in rats with a detection limit of 0.13 μM and a linear range of 0–80 μM.

In 2018, Gomathi et al. [95] took 4-methyl-2,6-bis (1,4,5-triphenyl-1H-imidazol-2-yl) phenol as fluorophore and 2,4-Dinitrophenyl ether (DNP) with strong electron-absorbing properties as the identification of H_2_S, and developed a H_2_S fluorescence sensor called DFI. Because of the strong electron absorption ability of DNP, it easily underwent nucleophilic substitution reaction. Therefore, DNP group easily enhanced the reaction rate and shorten the reaction time. The addition of H_2_S results in nucleophilic substitution reaction with DNP, resulting in the fracture of alkoxy bond and the formation of electron-absorbing groups. Afterwards, the excited-state intramolecular proton transfer process occurs and enhances the fluorescence. The fluorescence intensity of the sensor tended to be stable at 16 min, and the reaction time was greatly reduced compared with the above two sensors. However, the nucleophilic substitution reaction process was still slightly hindered due to the steric effect between fluorophore unit and DNP moiety. The sensor was used for selective detection of H_2_S in HeLa cells. The detection limit was as low as 0.13 nM and a narrow linear range of 0–10 μM was attained.

In addition, in 2021, Zheng et al. [96] synthesized Schiff base ISPBI by condensation reaction with *p*-phenylenediamine and 6-bromoisatin as the precursors, followed by N-alkylation with n-C_8_H_17_Br. The 3,3’-(1,4-Phenylenebis(azaneylylidene)) bis(6-bromo-1-octylindolin-2-one) (C8-ISPBI) colorimetric H_2_S sensor was obtained. In order to reduce the reaction time, C8-ISPBI used an isatin moiety, and this electron-deficient carbon was an active sensing site vulnerable to nucleophilic addition of H_2_S. The reaction time stabilized at 15 s, which was greatly reduced compared to the previous fluorescence and colorimetric H_2_S sensors. The color of the solution changed from light orange to dark pink after nucleophilic addition. C8-ISPBI also showed sensitization effects under practical conditions, such as film-coated paper strips and dyed wearable textiles, providing practical applicability for wearable devices.

The methods reported above are based on the improvement of the sensitive materials. Since organisms and living cells need the materials with good biocompatibility, low toxicity, and good biological activity, there are certain limitations in the design and synthesis of the desired materials. At present, the sensor applied in the detection of H_2_S has not proposed a better solution to the long nucleophilic reaction time. More endeavors are needed in this aspect in the future.

### 4.3. Test Material Preparation

When using dyes to detect H_2_S, the synthesis and preparation of dyes are often complicated. Especially for compound dye, the number of synthesis steps is greater, and the synthesis time is longer. If there is poor stability of the dye, it needs to be re-prepared before each test to address this problem.

In 2020, Noh et al. [97] used squaric acid 3 and N,N-dimethylaniline in one step to synthesize a new rectangular dye for selective detection of H_2_S. One-step synthesis greatly simplifies the preparation process. However, the synthesis process was a slightly longer. After adding the prepared dye into SiO_2_ and transferring the detection medium from the liquid phase to the solid phase, the experimental verification can be stored at room temperature for about 10 days, showing a certain stability. Similarly, Fan et al. [87] also prepared dansyl derivative (DBr) by one-step synthesis, but the synthesis process required stirring at room temperature for 24 h and drying for 4 h to complete the preparation.

In view of the complex problem of dye synthesis, no better method has been proposed at present. In most cases, one-step synthesis of dyes simplifies the steps in the preparation process, but the synthesis time is long and cannot show good stability.

## 5. Summary and Outlook

As one of the gas neurotransmitters, H_2_S affects many physiological and medical processes. Therefore, there is an increasing need for the detection of H_2_S in organisms and living cells with high sensitivity, selectivity, fast and reliable detection, and preferable visibility to the naked eye. Electrochemical sensors have high sensitivity, fast detection, and portability. Fluorescent and colorimetric chemical sensors can provide visual detection. Such chemical sensors provide a platform to solve the problem of H_2_S detection. Several kinds of chemical sensors applied to the detection of H_2_S are reviewed in detail based on the different properties of H_2_S, and the existing problems and possible solutions of these sensors are presented. The performance of H_2_S chemical sensors can be improved from the following aspects:Biocompatibility: Firstly, the detection materials with good biocompatibility and low toxicity should be applied. In addition, it can be combined with secondary substances with good biocompatibility, such as peptide-based substances, among others. Especially when detecting H_2_S via its metal affinity, metal ions are most likely to reduce cell activity or even cause toxicity to organisms. For this, precious metals, especially precious metal nanostructures such as gold nanoparticles/clusters, can be used, which have good biocompatibility. At present, a green and simple synthesis method has been proposed.Sensitivity: The detection technology with high sensitivity, such as ECL technology, or the combination of several detection technologies, such as PEC technology and EC technology, can be used to improve the sensitivity of the electrochemical sensor. Secondly, the detection materials could be optimized, such as nanomaterials with multi-dimensional structure, which have excellent chemical and optical properties and can provide a larger specific surface area to play the sensitization effect. Alternatively, to employ materials with higher conductivity is instrumental for the improvement of the sensitivity of electrochemical sensors.Reliability: Ratio detection, multi-channel detection, and multi-signal detection improve the reliability of sensors and ensure the accuracy of detection results by mutual evidence. With respect to the fluorescence chemical sensor, in order to solve the problem of false positives/false negatives, apart from the above methods, low background fluorescence interference and non-biological sample automatic fluorescence detection materials are preferable, such as UCNPs, PTMCs, etc.Response time: Detection methods based on the reducibility and nucleophilicity of H_2_S ultimately lie in the gain and loss of electrons. As a reducing agent and nucleophilic reagent for the loss of electrons, H_2_S can be introduced into the detection materials to reduce the response time of the sensor by introducing electron-absorbing groups, parent nuclear molecules, or multiple catalytic systems.

In addition, there are several problems and challenges with sensors used to detect H_2_S:Repeatability: H_2_S chemical sensors are mostly disposable, especially in fluorescence and colorimetric sensors, which cannot be reused. At present, almost all the repeatable chemical sensors are based on the affinity of H_2_S toward metal, which can be reused by repeated displacement or reversible reaction. However, the detection can only be repeated several times in a short period of time, which cannot guarantee the long-term stability, and the detection outcome becomes worse after multiple repetitions.Detection medium: At present, the detection of H_2_S is mostly in the liquid phase because of less interference under this condition. In order to realize the wearable H_2_S-detection device, it is urgent to transfer the detection medium to the solid phase. In this way, sensors can be combined with materials such as thin films and fibers. At the same time, it is also necessary to improve the detection selectivity to prevent interference of other gases except common anions and biological mercaptans in the solid phase.Detection limits: The amount of H_2_S in the human body is usually at the micromolar level, but in some systems, it is as low as the nanomolar level. Therefore, the detection limit of H_2_S sensor should be at least micromolar level, which can meet the detection in most cases. However, it is more necessary to develop sensors with detection limits of nanomolar and below so as to further realize the detection of very low content.Dye preparation: The problem of complex dye preparation often exists when dyes are used to detect H_2_S. The one-step synthesis method reduces the synthesis steps to a certain extent, but at the cost of long synthesis period wherein poor long-term stability of materials after synthesis always exists. More research should be carried out to this end.Real-time monitoring: Some H_2_S fluorescence and colorimetric chemical sensors possess long reaction time, which largely hinders the real-time monitoring. Therefore, it is very necessary to further improve the sensor sensitivity and shorten the response time.

In the process of summarizing the recent research advances of H_2_S chemical sensors for use in organisms and living cells, we realized that the research in this field is insufficient and many problems still need to be overcome. In H_2_S detection, the outstanding advantages of electrochemical sensors are high sensitivity and convenience. Fluorescent and colorimetric ones are characterized by visualization and suitable for in vivo imaging. If the sensor can be developed that combine with the advantages of the above chemical sensors, it will open up a new way for H_2_S detection and greatly improve the comprehensive performance of the H_2_S sensor. In the future, with the development and improvement of new nanomaterials, fluorescent dyes, wireless networks, and high-precision detection technologies, more new chemical sensors with favorable sensing performance will be designed to achieve real-time and visual detection to ensure the safety and health of physiological and pathological processes. Therefore, in this review, we hope to inspire readers to improve the method and open up new ways to further explore novel H_2_S chemical sensors with excellent properties.

## Figures and Tables

**Figure 1 sensors-23-03316-f001:**
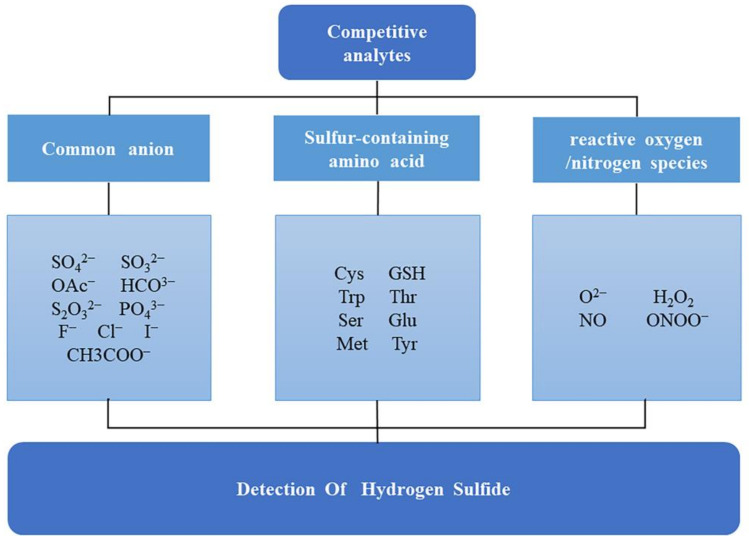
Common competitive analytes in organisms and living cells.

**Figure 2 sensors-23-03316-f002:**
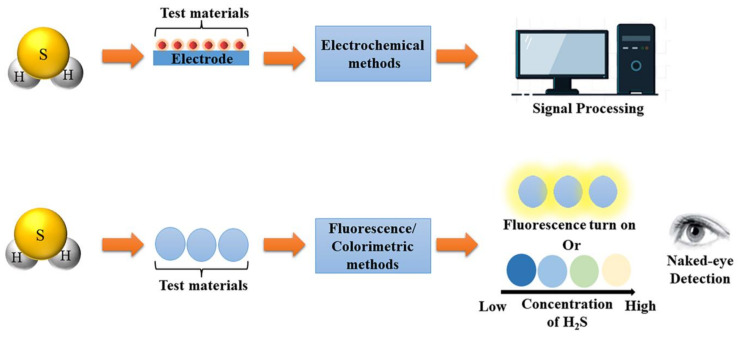
Diagram of chemical sensors for H_2_S detection.

**Figure 3 sensors-23-03316-f003:**
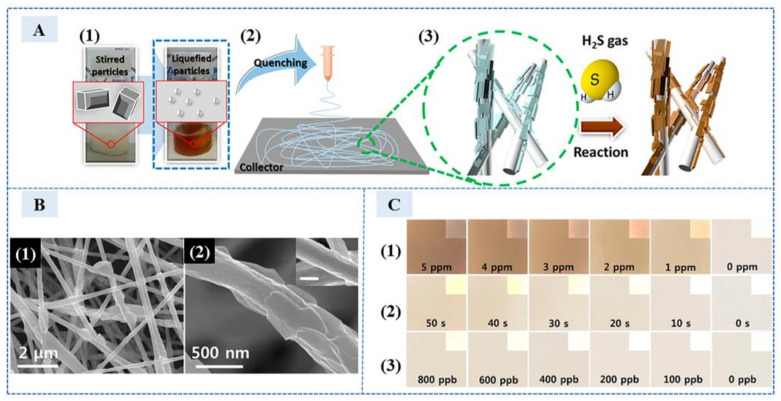
(**A**) Schematic illustrations of (1) high-temperature stirring (HTS), (2) electrospinning for synthesis of Pb(Ac)_2_@NFs, and (3) color change of Pb(Ac)_2_@NFs on exposure to H_2_S gas molecules. (**B**) Scanning electron microscopy (SEM) images of (1) HTS-Pb(Ac)_2_@NFs, (2) a strand of HTS-Pb(Ac)_2_@NFs (scale bar in the inset corresponds to 500 nm). (**C**) Photography images of HTS-Pb(Ac)_2_@NFs after (1) exposure to various concentrations (5−1 ppm) of H_2_S for 1 min, (2) exposure to 1 ppm of H_2_S for various exposure times (50−10 s), and (3) exposure to H_2_S in the concentration range from 800 to 100 ppb for 1 min (the insets indicate images modified to +20% lightness and −40% contrast). (Reprinted with permission from Ref. [24]. 2018, Cha et al.).

**Figure 4 sensors-23-03316-f004:**
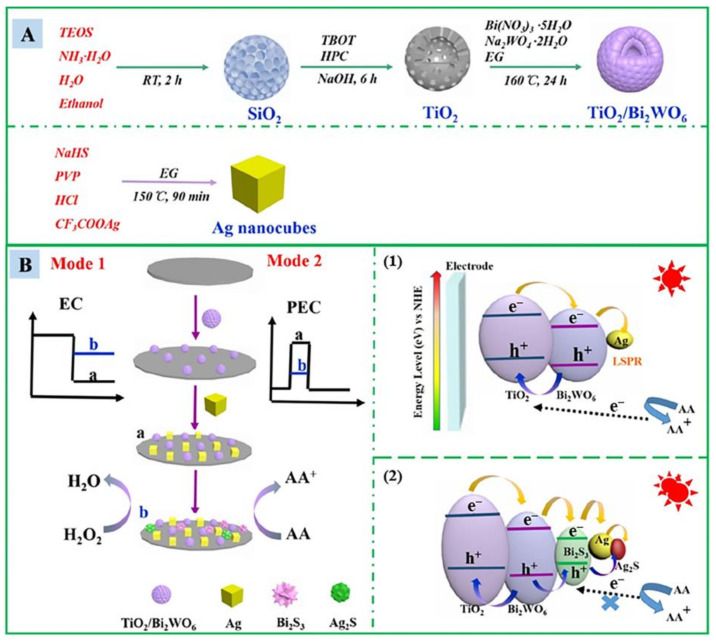
(**A**) Preparation process of TiO_2_/Bi_2_WO_6_ and Ag nanocubes; (**B**) Schematic illustration of the fabrication of dual-mode sensor for the H_2_S detection, and the proposed photocurrent signal on–off sensing mechanism based on the TiO_2_/Bi_2_WO_6_/Ag in the absence (1) and presence (2) of H_2_S. (Reprinted with permission from Ref. [31]. 2021, Xu et al.).

**Figure 5 sensors-23-03316-f005:**
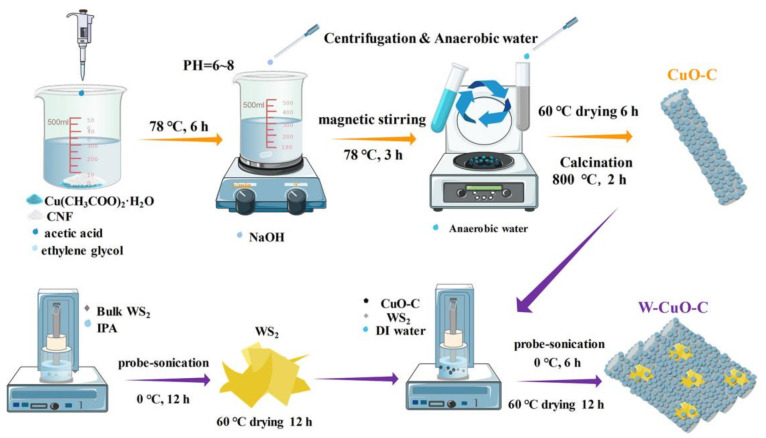
Schematic illustration of material preparation procedures. (Reprinted with permission from Ref. [34]. 2022, Li et al.).

**Figure 6 sensors-23-03316-f006:**
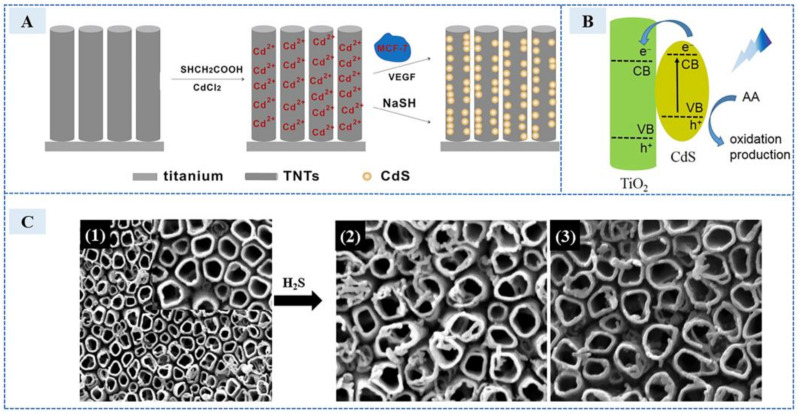
(**A**) The synthesis and detection process of the sensor. (**B**) Schematic illustration of photoelectrochemical process. (**C**) SEM images of (1) TiO_2_ nanotubes prepared by anodization at 30 V for 30 min, (2) CdS nanoparticles generation onto TiO_2_ nanotubes after treated with aqueous solution containing 1 mM NaSH (3) MCF-7 cells (10^8^ cells mL^−1^) with 40 ng/mL VEGF. (Reprinted with permission from Ref. [36]. 2016, Ding et al.).

**Figure 7 sensors-23-03316-f007:**
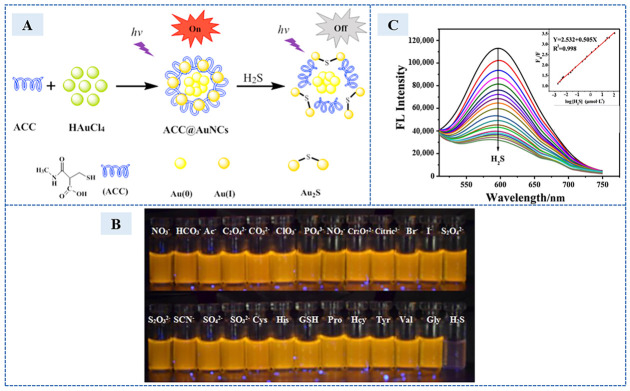
(**A**) The detection process of the fluorescent sensor. (**B**) Photographs of ACC@AuNCs with different analytes solution viewed under UV light at 365 nm. (**C**) Fluorescence responses of the sensors after the addition of H_2_S (from top to bottom: 0.002–120 μmol L^−1^). Inset: plot of the fluorescence peak intensity change (F_0_/F) versus the logarithmic of the concentration of H_2_S. (Reprinted with permission from Ref. [39]. 2017, Zhang et al.).

**Figure 8 sensors-23-03316-f008:**
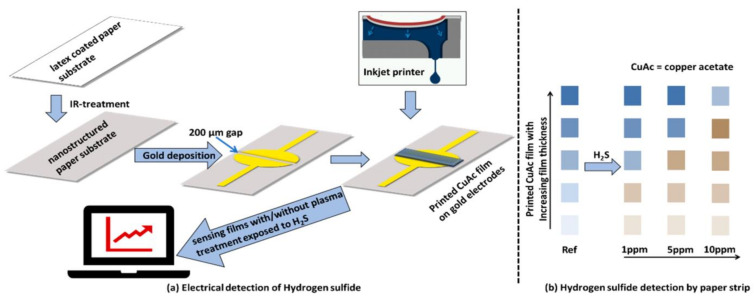
Schematic illustration of the development of gas sensing film on nano-structured paper substrate for electrical detection (**a**) and on specialty-paper substrate for colorimetric detection (**b**). (Reprinted with permission from Ref. [42]. 2019, Sarfraz et al.).

**Figure 9 sensors-23-03316-f009:**
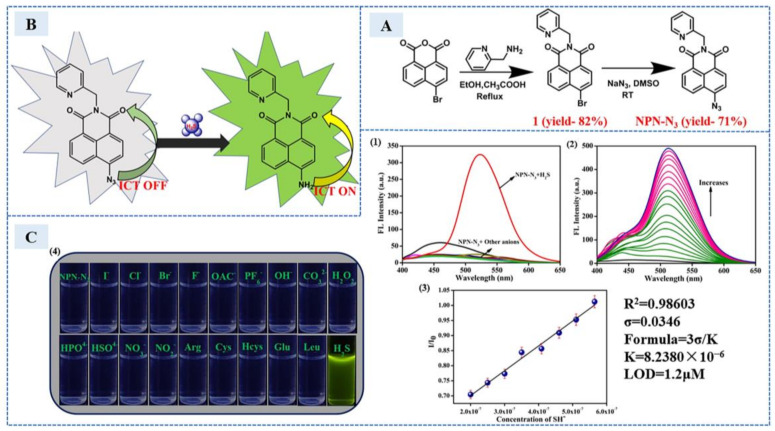
(**A**) The stepwise synthetic procedure of NPN-N_3_. (**B**) The emission response mechanism of NPN-N_3_ to H_2_S. (**C**) (1) Fluorescence spectra of the sensor NPN-N_3_ (20 μM) in the presence of different anions in acetonitrile/water (8:2 *v*/*v*), (2) NPN-N_3_ (20 μM) fluorescence spectra after progressive addition of H_2_S. The amounts of H_2_S were increased between 0 and 80 μM. (3) The linear relation of NPN-N_3_ for the concentration of H_2_S in the range of 0–10 μM. (4) The fluorescence color shift of NPN-N_3_ after adding different anions as seen under the UV light is depicted. (Reprinted with permission from Ref. [58]. 2022, Jothi et al.).

**Figure 10 sensors-23-03316-f010:**
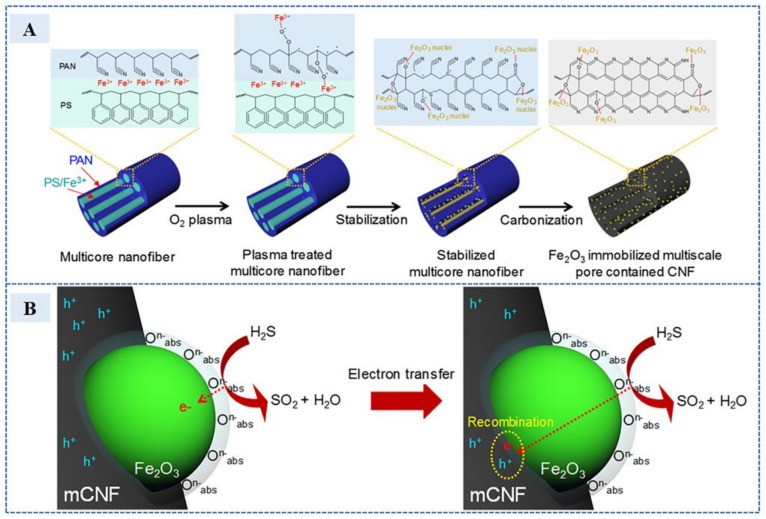
(**A**) A sequential composite diagram of Fe_2_O_3_ immobilized multiscale pore contained carbon nanofiber (Fe_2_O_3_-MPCNF). (**B**) Description of sensing and recovery mechanisms due to chemical reaction of H_2_S molecules and Fe_2_O_3_-MPCNF. (Reprinted with permission from Ref. [60]. 2022, Kim et al.).

**Figure 11 sensors-23-03316-f011:**
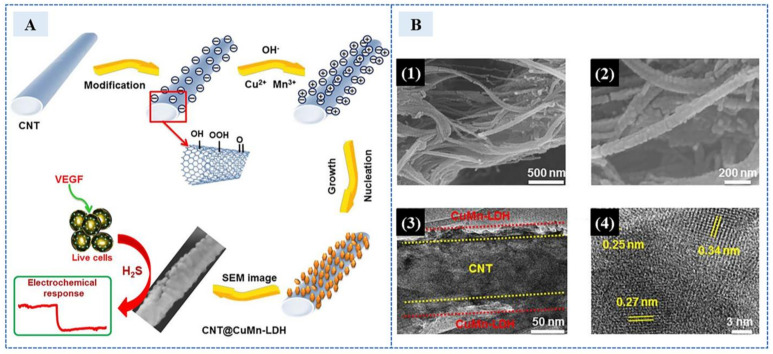
(**A**) Schematic illustration of various steps involved in the synthesis of hierarchical CNTs@CuMn-LDH nanohybrids and analytical performance for H_2_S detection. (**B**) (1,2) SEM images and (3) TEM image of CNTs@CuMn-LDH nanohybrids at different magnifications and (4) HRTEM image of CNTs@CuMn-LDH nanohybrids with lattice fringes of 0.25, 0.27, and 0.34 nm. (Reprinted with permission from Ref. [69]. 2019, Asif et al.).

**Figure 12 sensors-23-03316-f012:**
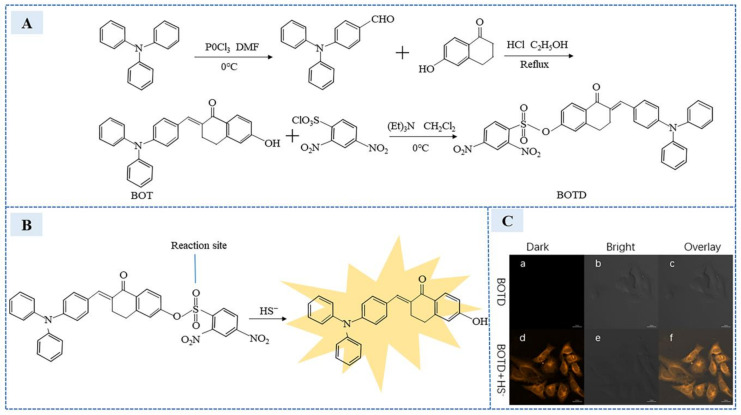
(**A**) Synthesis route of BOTD. (**B**) Schematic illustration of BOTD’s sensing towards HS^−^ cations. (**C**) Fluorescence images of HeLa cells (λex = 420 nm, λem = 500–550 nm): (left) dark field; (middle) bright field; and (right) overlay image. (a–c) Images of HeLa cells incubated with BOTD (10 μM/L) for 30 min; (d–f) fluorescence images of cells pre-treated with NaHS (100 μM/L) for 1 h and then incubated with BOTD (10 μM/L) for 30 min. (Reprinted with permission from Ref. [85]. 2020, Wang et al.).

**Figure 13 sensors-23-03316-f013:**
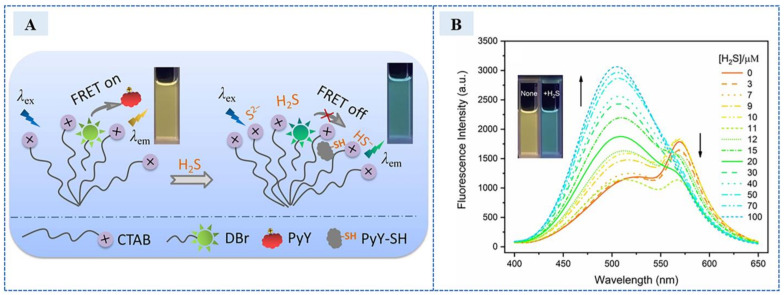
(**A**) Schematic representation of the dual-fluorophore sensor system for sensing H_2_S. (**B**) Fluorescence emission spectra of DBr/PyY/CTAB (10 µM/5 µM/100 µM) upon addition of H_2_S from 0 to 100 µM (Inset: Photos of the sensor platform before and after addition of 100 µM H_2_S under the illumination of 365 nm UV lamp). (Reprinted with permission from Ref. [87]. 2021, Fan et al.).

**Figure 14 sensors-23-03316-f014:**
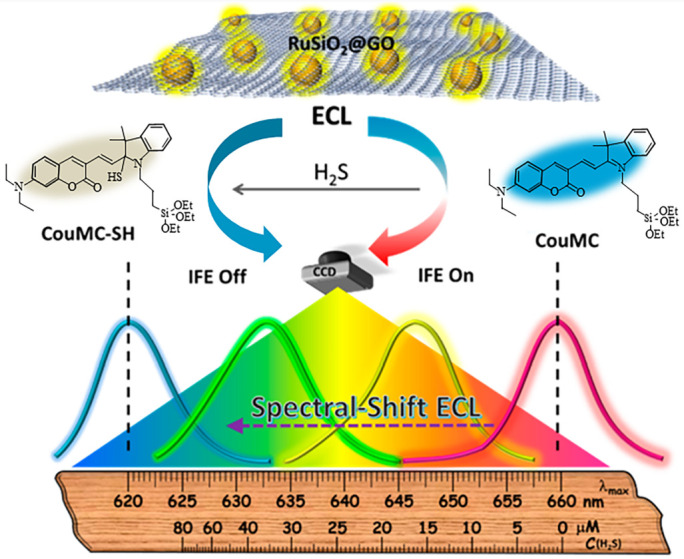
H_2_S sensing mechanism of CouMC and schematic diagram of the quantitative detection of H_2_S by measuring the *λ*max of the ECL spectra. (Reprinted with permission from Ref. [90]. 2018, Ma et al.).

**Figure 15 sensors-23-03316-f015:**
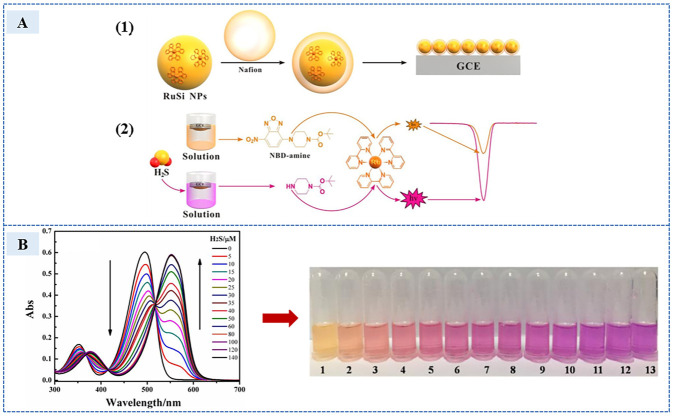
(**A**) Schematic diagram for fabrication of the ECL sensor. (1) The preparation of ECL sensor and (2) the reaction mechanism of ECL sensor. (**B**) UV–vis spectra of NBD-amine with different concentration of H_2_S, from 0 to 140 μM and visual color comparison of corresponding solution at different concentration of H_2_S, 1–13: 0, 5, 10, 15, 20, 25, 30, 35, 40, 50, 60, 80, and 100 μM. (Reprinted with permission from Ref. [93]. 2019, Huang et al.).

**Table 1 sensors-23-03316-t001:** Chemical sensors based on metal affinity.

Precipitation	Test Material	Linear Range	Limit of Detection	Response Time	Ref
CuS	NBD/Cu^2+^	0–20 μM	0.17 μM	—	[19]
CuS	[C_16_-2-C_16_im]Br_2_/Eu-POM/Cu^2+^	1.25–175 μM	1.25 μM	Seconds	[20]
CuS	naphthalimide–rhodamine B/Cu^2+^	0–40 mM/0–80 mM	0.23 mM/0.4 mM	<30 s	[21]
Ag_2_S	Au@Ag NPs	0.1–500 nM	0.04 nM	—	[22]
Ag_2_S	AgNPs	150–1000 ppbv	45 ppbv	10 min	[23]
PbS	Pb(Ac)_2_	—	400 ppb	1 min	[24]
ZnS	Mn-doped Zn^2+^ QDs	2–100 μM	0.3 μM	—	[25]

**Table 2 sensors-23-03316-t002:** Chemical sensors based on reducibility.

Reaction Type	Sensor Type	Test Material	Linear Range	Limit of Detection	Response Time	Ref
azide to amino	Fluorescent	AISA/poly	-	2.5 mM	-	[55]
azide to amino	Fluorescent	2,3-naphthalimide	0–80 μM	-	-	[56]
azide to amino	Fluorescent	4-azide-1,8-naphthalic anhydride/spiropyran	-	0.101 μM	25 min	[57]
azide to amino	Fluorescent	NPN-N3	-	1.2 μM	10 s	[58]
REDOX	Electrochemical	glassy carbon electrode	15 nM–15 μM	<100 nM	<10 s	[59]
REDOX	Electrochemical	Fe_2_O_3_-MPCNF	0.2–100 ppm	0.2 ppm	-	[60]

**Table 3 sensors-23-03316-t003:** Chemical sensors based on nucleophilicity.

Reaction Type	Test Material	Linear Range	Limit of Detection	Response Time	Ref
Nucleophilic addition	diethylaminocoumarin–hemicyanine	-	14 nM	-	[82]
Nucleophilic addition	3-hydroxyflavone	-	0.25 μM	<60 s	[83]
Thiolysis	coumarin derivatives/fluorobenzene	0–8 × 10^−6^ M/L	4 × 10^−6^ mol/L	-	[84]
Nucleophilic addition	BOTD	0–50 μM/L	27.3 nM/L	30 s	[85]

## Data Availability

Not applicable.

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
