# Peer review of "Review of Chemical Sensors for Hydrogen Sulfide Detection in Organisms and Living Cells"

_sensors, 2023, doi:10.3390/s23063316_

Round 1

Reviewer 1 Report

In the manuscript titled Review of Chemical Sensors for Hydrogen Sulfide Detection in Organisms and Living Cells, the authors categorized chemical sensors based on the different properties of hydrogen sulfide such as metal affinity, reducibility and nucleophilicity, and concluded recent progress and possible solutions. The logic is clear and the research investigation was convincing. A minor revision is suggested here due to the following issues to be well addressed prior to the formal publication.

1) The generalization of the optimal detection limit for hydrogen sulfide detection in organisms or living cells is not comprehensive.

2) Some phrases are organized inappropriately, such as “sensitive sensing platforms”. A lot of “in organisms and living cells”. The authors are suggested to confirm the specific application scenario first and avoid such repetitions later. “Based on metal affinity for H2S detection” should be replaced by “metal affinity based H2S detection”. Similar issues exist for the subtitle of 3 and 4.

3) The full name of some abbreviations is absent such as “LOD” in all tables. In addition, the definitions of some parameters such as response time and linear range should be added. And the unit of response time is not unified, sometimes “second” and sometimes “s”.

4) The images should be further improved as there exist original figure sequences probably to mislead the public readers such as Figs. 3, 4, 6, 9, 11, and 13.

Reviewer 2 Report

The authors have reported a review on chemical sensors for hydrogen sulfide (H2S) detection in organisms and living cells. They have systematically summarized the detection materials, methods, linear range, detection limits and selectivity of these sensors. It is a quality review with high importance. But I feel it still requires some changes before acceptance for publication as follows.

1. Thoroughly highlight some nanoparticle based sensors for H2S detection in living cells.

2. Highlight in details about research on the use of NIR fluorescent probes for this purpose.

3. Highlight some important research on the use of metal complex based probe for the detection of H2S in living cells.

Reviewer 3 Report

Yang and coworkers submitted the manuscript entitled “Review of Chemical Sensors for Hydrogen Sulfide Detection in Organisms and Living Cells”. In this review, the author breaks the usual review thinking and summarizes these H2S sensors from another perspective. First of all, the different mechanisms of hydrogen sulfide detection are classified. On this premise, the biocompatibility, sensitivity, accuracy, response time and other aspects are discussed. However, some aspects of the author's work, especially the details, need to be improved. I suggest you consider publishing this article after solving the following problems.

1.      The formatting of units in the article needs to be corrected.

1)            Spaces should be placed between data and units.

2)            Some units should be in italics. Such as micromoles and so on.

2.      Some of the illustrations are vague. For example, the chemical structure shown in Figure 12 is recommended to be drawn by yourself.

3.      The subcategories are out of order and difficult to compare when reading.

4.      Electrochemical sensor and fluorescence sensor have their own advantages. Is there a sensor that has all the advantages of both? The author should raise this question.

5.      In the summary and outlook, there is no explanation of how to review these chemical sensors in the article.

6.      The format of the reference needs to be corrected:

1)            Abbreviations or full names shall be uniformly used for journal names.

2)            The abbreviation of the journal name should be followed by a dot.

7.      The following references suggested to be cited:

1)            Analyst, 2022, 147, 4222–4227

2)            Chinese Chemical Letters, Volume 32, Issue 5, Pages 1799-1802

3)            Chinese Chemical Letters, Volume 32, Issue 8, Pages 2380-2384

Round 2

Reviewer 3 Report

Some interesting work (statement and refs)should be added such as:Org. Lett. 2019, 21, 5277−5280;Spectrochimica Acta Part A: Molecular and Biomolecular Spectroscopy 214 (2019) 355–359;Dyes and Pigments 165 (2019) 31–37 et al.
